# Amelioration of Behavioral Impairments and Neuropathology by Antiepileptic Drug Topiramate in a Transgenic Alzheimer’s Disease Model Mice, APP/PS1

**DOI:** 10.3390/ijms20123003

**Published:** 2019-06-19

**Authors:** Brice Ayissi Owona, Caroline Zug, Hermann J. Schluesener, Zhi-Yuan Zhang

**Affiliations:** Division of Immunopathology of the Nervous System, Institute of Pathology and Neuropathology, University of Tuebingen, Tuebingen D-72076, Germany; carolinezug@web.de (C.Z.); hermann.schluesener@uni-tuebingen.de (H.J.S.); zhiyuan.zhang@medizin.uni-tuebingen.de (Z.-Y.Z.)

**Keywords:** Alzheimer’s disease, APP/PS1 transgenic mouse, topiramate, amyloidosis

## Abstract

Alzheimer’s disease (AD) is a neurodegenerative disease that is the main cause of dementia in the elderly. The aggregation of β-amyloid peptides is one of the characterizing pathological changes of AD. Topiramate is an antiepileptic drug, which in addition, is used in the treatment of many neuropsychiatric disorders. In this study, the therapeutic effects of topiramate were investigated in a transgenic mouse model of cerebral amyloidosis (APP/PS1 mice). Before, during, and after topiramate treatment, behavioral tests were performed. Following a treatment period of 21 days, topiramate significantly ameliorated deficits in nest-constructing capability as well as in social interaction. Thereafter, brain sections of mice were analyzed, and a significant attenuation of microglial activation as well as β-amyloid deposition was observed in sections from topiramate-treated APP/PS1 mice. Therefore, topiramate could be considered as a promising drug in the treatment of human AD.

## 1. Introduction

Alzheimer’s disease (AD) is a neurological disorder and the main cause of dementia in the elderly. The disease is characterized by memory loss and the deposition of β-amyloid plaques and neurofibrillary tangles (NFTs) in the brain [1]. Impairments in cognitive ability and social interaction caused by neurotoxicity and neuroinflammation are associated with AD. Amyloid plaques are extracellular deposits of amyloid beta (Aβ) in the grey matter of AD brains, which is one of the characterizing pathological changes [2].

Patients with AD have an increased risk of developing seizures or epilepsy [3]. It was reported that generalized and unprovoked seizures occur late in the course of AD, and patients with an early onset of AD are particularly susceptible to develop seizures [4]. For this reason, the use of antiepileptic drugs (AED) for patients with higher brain dysfunction has been analyzed in several clinical studies [3]. For instance, the anticonvulsant and mood-stabilizing agent valproic acid (VPA) has been widely used for the treatment of epilepsy. VPA was found to decrease β-amyloid production and aggregation through an inhibition of GSK-3beta-mediated gamma secretase cleavage of APP in vitro and in vivo [5]. Many other potential antiepileptic agents have been tested acting on various mechanisms of epilepsy. Topiramate (Figure 1) is a Food and Drug Administration (FDA)-approved drug that is frequently prescribed for epilepsy and the prevention of migraines. It is also an AED, which is used to slow down the symptoms of AD that have been shown to be effective in the treatment of many neuropsychiatric disorders.

Several transgenic mouse models are available to evaluate memory loss, behavioral deficits, and social interaction ability, all of which are associated with cerebral amyloidosis. Transgenic animal models can reconstruct cellular, biochemical, and molecular alterations associated with human diseases, and may help identify key disease mechanisms [6]. For instance, behavioral deficits could be ameliorated after the treatment of APP/PS1, JNLP3, and APP23 mice with cannabidiol and oridonin [7,8]. In APP/PS1 mice, amyloid plaque deposition starts at approximately six weeks of age in the neocortex. Deposits appear in the hippocampus at about three to four months and in the striatum, thalamus, and brainstem at four to five months. A study published a few years ago reported the effect of topiramate in combination with levetiracetam on APPswe/PS1dE9 mice after 30 days of treatment. In our study, we used for a limited time topiramate alone in an APP/PS1-21 transgenic mouse model, and reported its effects on social behavior and the amelioration of disease pathology.

## 2. Results

### 2.1. Effect of Topiramate Treatment on Social Interaction

At the beginning of our experimental treatments and after 10 days, no significant differences regarding interactive behaviors were observed between APP/PS1 mice treated with topiramate and the non-treated controls. However, after 21 days of treatment, topiramate restored the impairment observed in non-treated mice by significantly increasing the frequency of interactive behavior (control = 5.571 ± 1.631, topiramate = 12.00 ± 1.543, *p* < 0.05, *n* = 6) (Figure 2A). No significant effect was observed in the distance traveled between the controls and topiramate-treated mice (Figure 2B), which means that the motor function was not damaged.

### 2.2. Effect of Topiramate Treatment on Behavioral Impairment (Nest Construction Assay)

APP/PS1 mice were further tested for the effect of topiramate treatment on affiliative behavior. We observed an impairment of nesting capacity in non-treated APP/PS1 mice (previously published data) [8]. On the first day of treatment with topiramate (Day 1), no significant difference was observed between treated and non-treated mice (control = 1.143 ± 0.184, topiramate = 1.357 ± 0.170, *n* = 6) (Figure 3A). We observed the same result after 11 days of treatment with topiramate (control = 1.500 ± 0.154, topiramate = 1.857 ± 0.142, *n* = 6) (Figure 3B). However, after 21 days of treatment, a significant difference between treated and non-treated mice was observable (control = 1.286 ± 0.1844, topiramate = 1.929 ± 0.1700, *p* < 0.05, *n* = 6) (Figure 3C), by an immediate chewing and tearing of the paper towel fragments, which were tidily torn into pieces and grouped into a corner of the cage. In contrast, APP/PS1 transgenic mice from the non-treated group slightly chewed the paper towels, with no real destruction observed, and they were rather found littered all over the cage.

After this restoration of behavioral function by topiramate, the effects regarding neuropathological changes and neuroinflammation were studied.

### 2.3. Effects of Topiramate on Neuroinflammation and Amyloidosis

We did not observe any significant difference in amyloid-β deposition or in Iba-1 expression between APP/PS1 mice from the control group and their littermates (receiving no treatment). Topiramate oral treatment mitigated neuropathological changes, in comparison to the non-treated mice. Topiramate significantly reduced plaque number in the cortex (control = 144.1 ± 9.660, topiramate = 98.00 ± 4.488, *p* < 0.05, *n* = 6) and hippocampus (control = 13.86 ± 2.098, topiramate = 6.286 ± 0.8650, *p* < 0.05, *n* = 6) (Figure 4A,C). Topiramate also significantly reduced the Aβ immunoreactivity (IR) area in both the cortex and hippocampus (cortex: control = 0.7192 ± 0.022, topiramate = 0.4054 ± 0.0097, *p* < 0.05; hippocampus: control =0.1883 ± 0.014, topiramate = 0.1243 ± 0.016, *p* < 0.05, *n* = 6) (Figure 4B,D). Moreover, in brain sections of mice treated with topiramate, Aβ plaques were of smaller size and had fewer branches (Figure 5B) in comparison to control mice (Figure 5A).

Topiramate also significantly reduced the IR area of Iba-1+ cells in the cortex and the hippocampus (cortex: control =0.400 ± 0.024, topiramate = 0.3071 ± 0.0095, *p* < 0.05, hippocampus: control = 0.2929 ± 0.03, topiramate = 0.1014 ± 0.04, *n* = 6) (Figure 4E,F). We observed only very few cells expressing Iba-1, and the majority of them were also less clustered around amyloid plaques in the cortex after topiramate treatment, indicating a reduction of microglial activation (Figure 5).

## 3. Discussion

Treating behavioral and memory loss symptoms that characterize dementia remains a difficult task. Here, we describe the effect of the second-generation antiepileptic drug topiramate in a APP/PS1 mouse model of AD. After 21 days of treatment, topiramate restored the capacity of mice to construct nests as well as their social interaction behavior. Immunohistochemical results indicate that the oral treatment of topiramate significantly reduced the deposition of β-amyloid and the activation of microglial cells in the brain of APP/PS1 mice. The treatment of wild-type mice (healthy) with topiramate showed no effect of toxicity. Oral treatment with topiramate showed protective effects against behavioral impairments and neuropathological changes in APP/PS1 transgenic mice. Topiramate significantly ameliorated behavioral deficits by restoring impaired nesting behavior and the social interaction of transgenic mice, possibly by both attenuating the deposition and the aggregation of β-amyloid as well as the activation of microglia in the brains of the mice (cortex and hippocampus).

Social interaction and non-cognitive deficits are primary indicators of AD [9]. Very early on, 50% of AD patients exhibit depression symptoms with a concomitant paucity of social behavior and memory recall [10]. APP/PS1 transgenic mice were used in this study because they exhibit a relatively early deposition of amyloid plaques at two months in the cortex, and two months later in the hippocampus [11]. Cognitive impairments in animal models are normally tested by many other methods, including the Morris water maze [12]. However, the symptoms of cognitive impairments appear rather only at eight months of age [13], and therefore could not be used to analyze the potential of treatments at an early stage of the disease. A study on topiramate showed that, in combination with leveraticetam, topiramate alleviates behavioral deficits and reduced amyloid plaque formation in seven-month-old APPswe/PS1dE9. Therefore, our study differs from this one in many aspects. We have used five-month-old APP/PS1 mice to mimic the earlier symptoms of the disease, and a relatively short period of treatment (21 days) in comparison to 30 days of treatment with a combination of topiramate and leveraticetam. To evaluate the effect of treatment on AD transgenic mice models, nest construction and social interaction assays were used [14,15]. Many studies reported in the literature have mentioned its effects on epilepsy and many other disorders [16]. Topiramate treatment (30 days) was reported to prevent the apoptosis of hippocampus neurons in adult Wistar rats that received an injection of Aβ-40 into the hippocampus [17]. Topiramate (25–50 mg/d) had a similar efficacy with risperidone in controlling behavioral disturbances in patients with dementia [18]. Topiramate also showed positive effects on serum leptin, body mass index, and fasting insulin-to-glucose ratio levels in prepubertal children [19]. Behavioral impairment and loss of memory remain serious hallmarks of AD patients. The effect obtained with topiramate in an APP/PS1 transgenic mouse model for AD suggests that it could be a promising lead compound for use in clinical studies for patients suffering from AD.

Many scientific reports have demonstrated that epileptic seizures and AD are somehow related. For instance, valproic acid, a first-line AED, exerted protective effects in mouse models of AD [20]. Many other anticonvulsant mood stabilizers have shown beneficial effects in the treatment of behavioral and psychological symptoms of dementia [21]. However, it remains unclear whether new AED could reverse neuropathology in AD transgenic mice models. Topiramate has been used successfully as a treatment for alcohol dependence [22], methamphetamine addiction [23], cocaine addiction [24,25], obesity [26], and antipsychotic-induced weight gain [27]. It has been found to be increasingly effective for migraine sufferers with limited side effects [28,29]. However, it is noteworthy that some side effects of the drug have been reported. For instance, patients under topiramate medication have frequently reported an impairment of attention as a side effect of the drug, as well as a frequent complaint of epilepsy [30]. Some rare cases of treatment with topiramate were reported to induce acute onset myopia during use for migraines [31].

Topiramate was described to be a potential histone deacethylase (HDAC) inhibitor. Due to their neuroprotective and anti-inflammatory effects, HDAC inhibitors have been regarded as a promising therapy for neurodegenerative diseases. However, drugs used for the treatment of epilepsy may not reverse cognitive deficits in APP/PS1 models of transgenic mice. Topiramate is widely prescribed in many neurological disorders, and is reported to be effective at reducing seizure frequency, and also does not cause adverse effects in comparison with old AEDs [32]. The impairment of attention is frequently reported as one of the main side effect of antiepileptic medication, and the beneficial effect of topiramate on attention has been reported in human patients [33]. Apart from topiramate, cannabidiol has been shown to have potential effect as a preventative treatment for AD with a particular relevance for symptoms of social withdrawal and facial recognition [7]. Moreover, it was shown that topiramate prevents the apoptosis of hippocampus neurons, by increasing the expression of Bcl2 and a surviving and decreasing expression of Fas, Bax, and caspase-3 in Wistar rats [34].

Neurodegenerative diseases of the central nervous system are most often characterized by cognitive deficits and non-mnemonic behaviors. The APP molecule that is produced during brain degeneration as well as the products of inflammatory reaction are reported to be directly associated with deficits in social interaction, memory loss, and behavior [35]. Therefore, the APP/PS1 model of transgenic mice is valuable for modeling cognitive impairment in AD. The capacity to construct nests is a social behavior that is important for mice survival. Indeed, an impairment of nest-construction capacity has been reported in a similar Tg2576 mice model [36]. In the model of APP/PS1 mice for AD used in this study, we observed a similar impairment of nesting construction capacity, compared with the non-transgenic mice. After treatment, topiramate could significantly restore the impaired nest-construction capacity (Figure 3C). The flavone hesperidin was also found to significantly restore deficits in non-cognitive nesting ability and the social interaction of APP/PS1 mice after relatively short-term treatment [37]. Deficits in social communication is one of the characteristics of Alzheimer’s disease. Impairment in social interaction was reported in a similar model of APP/PS1 mice [38]. APP/PS1 double transgenic mice also demonstrated an improvement in memory loss and social behavior after they were cohoused with wild mice [8]. Another study showed that non-cognitive AD symptoms such as deficits in social interest, interaction, and communication are expressed very early in APP and APP-PS1 mice, suggesting that social deficits appear before cognitive symptoms in AD [39]. In this study, we observed very similar deficits in APP mice social interaction, in comparison with non-transgenic mice. After 21 days of treatment with topiramate, the impairment in social interaction was significantly ameliorated (Figure 2A). Topiramate combined with levetiracetam also induced protective effects against the impairment of behavior and β-amyloid plaque formation and deposition in the brains of APPswe/PS1dE9 transgenic mice [20]. The amelioration of behavioral deficits as well as the reduction of Aβ plaque deposition may be related to the reduction of inflammation in the brain.

It has been reported that APP/PS1 transgenic mice show epileptic discharges from a young age and exhibit β-amyloid plaque deposition. However, the mechanism of action of topiramate on neuroinflammation has not been clearly elucidated. In this study, anti-inflammatory results obtained with topiramate on N9 microglial cells showed no significant effect, suggesting that topiramate may exert its protective effects on the brain through many other different pathways such as the inhibition of HDAC; furthermore, topiramate was described as an HDAC inhibitor [40]. A study of topiramate on an animal model of epilepsy reported that the neuroprotective effect of TPM seems to be related to its capacity to inhibit mitochondrial permeability [41].

From the results obtained in this study, reduced Aβ deposition, especially plaque-associated APP density, are important factors for the improvement of nest-constructing capacity and social interaction in these mice. Taken together, treatment with topiramate effectively ameliorated nesting behavior, social interaction, and neuropathology in an APP/PS1 model of transgenic mice. The beneficial effects of topiramate observed in this study could be explained by multiple factors including decreased Aβ deposition and APP expression, as well as the possible immunoregulation of microglia inflammation in the brain. All these results suggest that the antiepileptic drug topiramate could be a promising therapeutic drug for the treatment of human neurodegenerative diseases.

## 4. Materials and Methods

### 4.1. Animals

Male APP/PS1-21 mice were obtained from Professor Jucker (Hertie-Institute, Tuebingen, Germany). Heterozygous male APP/PS1-21 mice were bred with wild-type C57BL/6J females (Charles River Germany, Sulzfeld, Germany). Offspring were tail-snipped and genotyped using PCR with primers specific for the APP-sequence (Forward:’ GAATTCCGACATGACTCAGG’, Reverse: ‘GTTCTGCTGCATCTTGGACA’). All the experiments were approved by the institutional animal care committee of Tuebingen University and conducted according to the German Animal Welfare Act (TierSchG) of 2006. Ethic approval code: HF02/11, date: February 2012, approval name of the ethics committee: Regierungspräsidium Tübingen (Regional Council).

### 4.2. Materials

Topiramate was purchased from Carbosynth Ltd. (Compton, Berkshire, UK). For oral treatment, topiramate was suspended in 1% carboxymethylcellulose (CMC, Blanose, Hercules-Aqualon, Düsseldorf, Germany) and administered by gavage at a dose of 20 mg/kg. Control mice received an equivalent volume of CMC.

### 4.3. Treatment with Topiramate

Two groups of transgenic mice (*n* = 6, three males and three females at the age of 5 months) were used in this study. The treatment was applied for 21 days: 1. Topiramate treatment (daily, 20 mg/kg bodyweight), 2. Vehicle-treated used as control.

### 4.4. Social Interaction Assay

We have performed the social interaction assay according to previous protocols with minor modifications [9]. The experiment was recorded in order to monitor all the distinct behaviors from control and topiramate-treated APP/PS1 mice (referred to as the resident in this assay) in the presence and absence of another mouse (referred to as the intruder). Each mouse was placed in a clean plastic cage and allowed to accommodate for 15 min (resident mouse). Thereafter, an age, weight, and gender-matched and non-treated naïve mouse referred to as the intruder mouse was introduced into the cage for another 15 min. The 30-min experiments were videotaped, and identifiable independent and interactive behaviors were recorded by three independent observers who were blinded to the different treatments. The resident mouse was allowed to spend 15 min alone and 15 min with the intruder, while the intruder mouse stayed with the resident mouse for 15 min.

### 4.5. Design and Evaluation of Nest Construction Assay

A nest construction assay was modified to determine the deficits in affiliative/social behavior of APP/PS1 mice and potential changes following treatment. For this purpose, different mice treated with topiramate or control were housed in clean plastic cages for 24 hours. At the same time, 1 cm of wood chip beddings were introduced into the cages. In order to monitor the mice nest-construction capacity, each cage was supplied with a paper towel that had been turned into 5 × 5 cm square pieces. The next morning, individual cages were inspected, and the next construction was scored using a three-point system as reported in our previous studies [8].

### 4.6. Immunohistochemistry and Image Evaluation/Analysis

Topiramate-treated and control mice were sacrificed after 21 days of treatment. They were perfused intracardially with 4 °C, 4% paraformaldehyde in PBS (phosphate buffer saline). Brains were quickly removed and post-fixed in 4% paraformaldehyde overnight at 4 °C. Brains were then cut into two hemispheres, embedded in paraffin, serially sectioned (3 µm), and mounted on silan-covered slides. Hemispheres sections were stained by immunohistochemistry as described previously, with the following antibodies: β-amyloid (1:100; Abcam, Cambridge, UK) for Aβ deposition, and Iba-1 (1:200; Wako, Neuss, Germany) for activated microglia. For data analysis, hemisphere sections were examined using a Nikon Coolscope light microscope. Two independent observers randomly numbered and analyzed the different sections. The total number of Aβ plaques and microglia cells positive for Iba-1 staining in the neocortex and hippocampus was calculated. Images of cross-sections were captured and areas of interest (neocortex and hippocampus) were outlined and analyzed. Furthermore, area percentages of specific immunoreactivity (IR) were also calculated.

### 4.7. Statistical Analysis

Differences of plaque/cell counts, area percentages, and behavioral data were analyzed by unpaired *t*-tests (Graph Pad Prism 6.0 software). For all the statistical analyses, significance levels were set at *p* < 0.05.

## Figures and Tables

**Figure 1 ijms-20-03003-f001:**
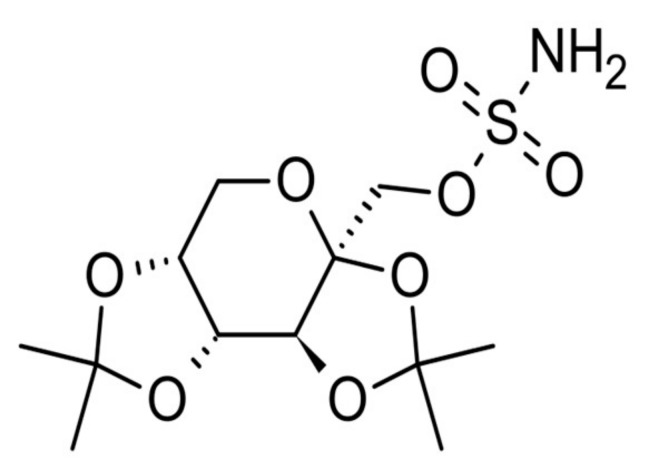
Molecular structure of topiramate.

**Figure 2 ijms-20-03003-f002:**
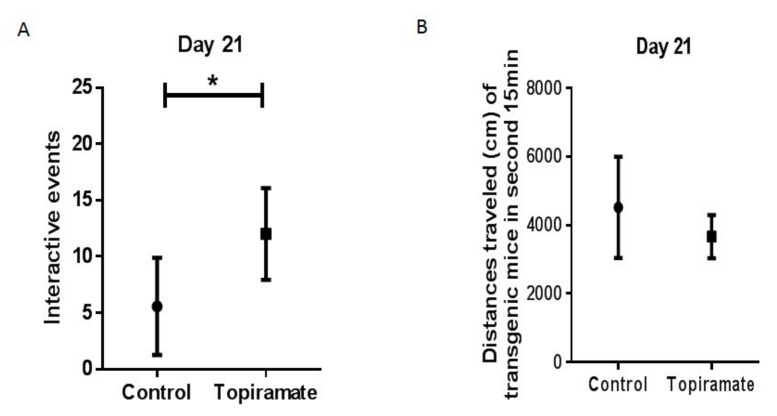
Effect of topiramate treatment on behavioral impairments (social interaction assay). APP/PS1 mice received oral treatment of topiramate for 21 days. Social interaction was determined by a resident-intruder assay (see Materials and methods). (**A**) As baseline controls of social interaction, APP/PS1 mice had less interactive behavioral events and more independent behavioral events together with introduced intruder mice compared with naïve mice. Following treatment, increased behavioral events were observed in the topiramate group in comparison with the controls. (**B**) We observed no significant change in the distance traveled both in the control and topiramate-treated group.

**Figure 3 ijms-20-03003-f003:**
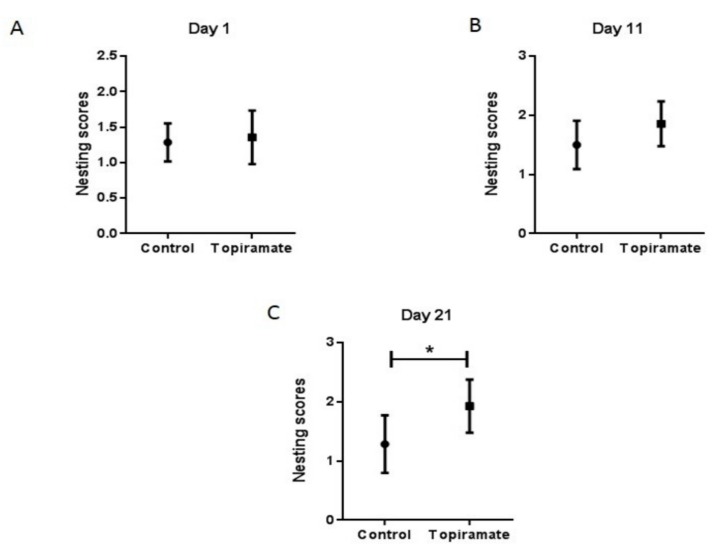
Effect of topiramate treatment on behavioral impairments (nest construction assay). APP/PS1 mice received oral treatment of topiramate for 21 days. Together with naïve mice, they were assessed for nesting behavior. Nest construction was explored with paper towel material using a three-point scaling system (see Materials and methods) in naïve and APP/PS1 mice. (**A**) No significant difference between the topiramate and the control group could be observed right at the beginning of treatment, namely at Day 1. (**B**) At Day 11, a non-significant increase of the nesting score was observed in the topiramate-treated group. (**C**) A significant difference between the topiramate and control groups was observed after 21 days of treatment (*p* < 0.05).

**Figure 4 ijms-20-03003-f004:**
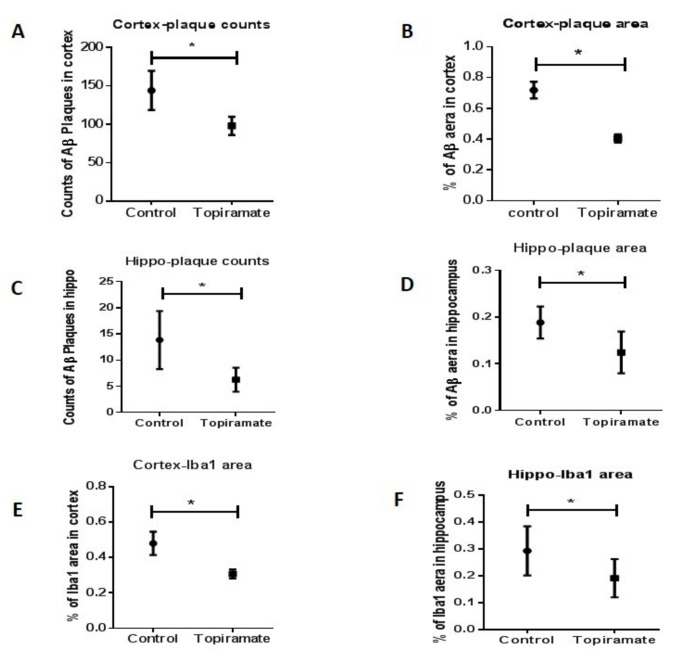
Effect of topiramate on β-amyloid deposition and microglial activation. APP/PS1 were either treated with topiramate or carboxymethylcellulose (CMC) (used as vehicle for control group). After 21 days of treatment by gavage, brains from transgenic mice and their untreated littermates were analyzed by immunohistochemistry. Cells and plaque count as well as arithmetic means of the immunoreactivity (IR) area are represented in different graphs. The unpaired t-test was used to calculate the differences of plaque/cell counts and area percentages between the treatment group and control, with significance levels set at *p* < 0.05. (**A**) Topiramate significantly reduced the number of amyloid plaques in the cortex of APP/PS1. (**C**) A reduction of plaque number was also observed in the hippocampus of mice. (**B**,**D**) IR percentages of amyloid plaques were significantly reduced in both the hippocampus and cortex after topiramate treatment. (**E**,**F**) In APP/PS1 mice treated with topiramate, the Iba-1 IR was significantly reduced in both the cortex and hippocampus.

**Figure 5 ijms-20-03003-f005:**
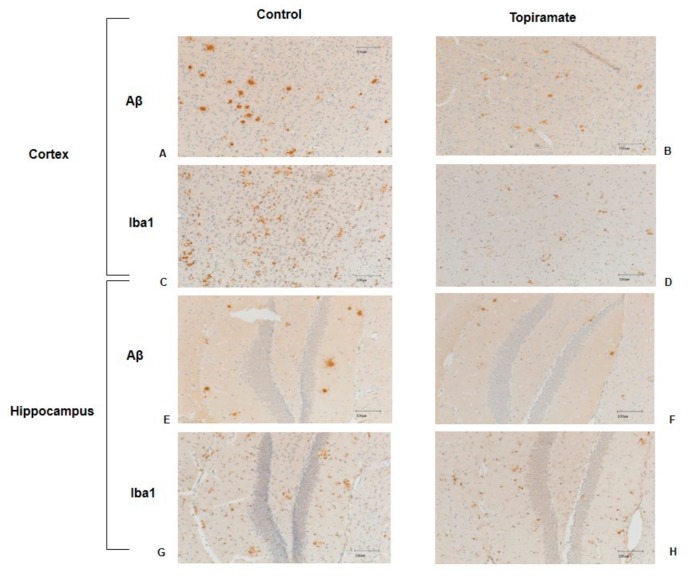
Effect of topiramate on amyloid beta (Aβ) accumulation and inflammation activation. Images show the deposition of Aβ and microglial activation in different treatment groups. (**A**,**B**) APP/PS1 mice from the control group showed larger Aβ plaques in the cortex (**A**) in comparison to the topiramate-treated group (**B**). (**C**,**D**) Microglia staining for Iba-1 showed higher cell numbers and a larger IR area of microglia surrounding the plaques in the cortex of the control mice. (**E**,**F**) The hippocampus of non-treated mice showed larger Aβ plaques in comparison to treatment (topiramate). (**G**,**H**) Results from microglia activation by targeting Iba-1 showed that the cells were clustered around amyloid plaques and distributed throughout the hippocampus in both the control and topiramate groups.

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
