# Peer review of "Amelioration of Behavioral Impairments and Neuropathology by Antiepileptic Drug Topiramate in a Transgenic Alzheimer’s Disease Model Mice, APP/PS1"

_ijms, 2019, doi:10.3390/ijms20123003_

Round 1

Reviewer 1 Report

Manuscript ID: ijms-502612: Amelioration of behavioral impairments and neuropathology by antiepileptic drug Topiramate in a transgenic mice model of cerebral amyloidosis

In general, there are many reports that epilepsy is deeply related to dementia, including not only Alzheimer's disease but other types of dementia. From this point of view, the present report is very timely and important. This manuscript is worth being published in the Journal for the first step of the development of the drugs for dementia by drug repositioning, from epilepsy to dementia. The experiments were well-performed and the manuscript is almost ready for the publication. There are some suggestions which may improve the value of this report.

<Major Points>

(1) The effect of Topiramate on wild type mice.

The observed effect of topiramate may not be specific to pathological mice. Its effect on wild type (healthy) mice should be shown in at least one assay or discussed.

(2) Vehicle control vs control

There is a description of the vehicle-treated control (line 246), the data were shown by the comparison between topiramate-treated mice and naive mice. The vehicle-treated mice should be used as control. This should be explained.

(3)The dosage of topiramate

The clinical daily dosage of topiramate in adult is around 100mg. That means around 2 mg/kg. Hence, the dosage of 20 mg/kg used in the present report may be clinically too high. This discrepancy should be discussed. Also the reason why this dosage was used in the present report should be described.

<minor points>

(a) The numbering of Figs. Fig 2 and Fig 3 should be exchanged.

In line 65 there is Fig 3 followed by Fig 2 in line 88. This is very unusual.

(b) Fig. 1 should be omitted.

Topiramate is well-known pharmaceutical substance and its molecular structure is not important in the present report. In addition, Fig 1 is not referred to in the text.

(c) The reference of "(previously published data)" in line 75 is necessary.

Line 75 to 76. "We observed an impairment of nesting capacity in non-treated APP/PS1 mice (previously published data)." The reference of the previous data is necessary.

(d) The method of Topiramate administration (line 243, Treatment with Topiramate).

Although Topiramate seems to be administered by orally, it should be described. The detail how it was orally administered should also be described.

(e) Animal care. (line 232, 4,1 Animals). 

The admittance of the animal experiments by the institutional animal care committee or the description of the compliance to the general guide lines of the care and use of laboratory animals is necessary.

(f) Title should include "Alzheimer’s disease".

APP/PS1 mice is generally regarded as Alzheimer model mice. One suggestion of the title is as follows. "Amelioration of behavioral impairments and neuropathology by antiepileptic drug Topiramate in a transgenic Alzheimer’s disease model mice, APP-PS1.

"cerebral amyloidosis" may include vascular amyloidosis.

Author Response

Response to reviewers

Manuscript IJMS- 50612

Dear Dr Josephine Xu and Editors,

Thank you very much for consideration of a new version of our manuscript entitled " Amelioration of behavioral impairments and neuropathology by antiepileptic drug Topiramate in a transgenic mouse model of cerebral amyloidosi". We would also like to thank the reviewers for the many helpful comments and suggestions.

In the following, we would like to give a point-by-point response to the comments of the reviewers. In the revised manuscript, we have also marked all the changes.

Reviewer #1:

(1) The effect of Topiramate on wild type mice.

The observed effect of topiramate may not be specific to pathological mice. Its effect on wild type (healthy) mice should be shown in at least one assay or discussed.

Response: We appreciate the reviewer’s comment on this issue. The effect of topiramate on wild type mice was investigated in a third group of treatment with data not included in this article since no effect or toxicity of the drug was observed. We were therefore interested in using the vehicle-treated transgenic mice with pathological defects as a control. This is the reason why we mentioned 3 group of treatment in the material and methods section in the beginning. We have included a sentence in the discussion to point it out. Please see line 145.

(2) Vehicle control vs control

There is a description of the vehicle-treated control (line 246), the data were shown by the comparison between topiramate-treated mice and naive mice. The vehicle-treated mice should be used as control. This should be explained.

Response: The reviewer is right on this point. The confusion has been clarified accordingly. We have described two groups of treatment: The vehicle-treated group and the Topiramate-treated group. Please see line 254.

(3) The dosage of topiramate

The clinical daily dosage of topiramate in adult is around 100mg. That means around 2 mg/kg. Hence, the dosage of 20 mg/kg used in the present report may be clinically too high. This discrepancy should be discussed. Also, the reason why this dosage was used in the present report should be described.

Response: After preliminary toxicity studies with topiramate on mice, we have calculated the LD50 (100mg/kg) to estimate the maximum dose which is safe to use. Then, 20mg/kg was a dose selected below the LD25 and was used for in vivo studies. Moreover, many reports on in vivo studies in oral administration with topiramate on mice used up to 50mg/kg, in our case, 20mg/kg was the most effective dose.

<minor points>

(a) The numbering of Figs. Fig 2 and Fig 3 should be exchanged.

In line 65 there is Fig 3 followed by Fig 2 in line 88. This is very unusual.

Response: We thank the reviewer for this observation. The error may have occurred during the editing process. However, the numbering has been exchanged in the manuscript.

(b) Fig. 1 should be omitted.

Topiramate is well-known pharmaceutical substance and its molecular structure is not important in the present report. In addition, Fig 1 is not referred to in the text.

Response: Figure 1 has been referred to in the manuscript. We would like to present Topiramate’s molecular structure to the reader and therefore suggest keeping figure 1 in the manuscript.

(c) The reference of "(previously published data)" in line 75 is necessary.

Line 75 to 76. "We observed an impairment of nesting capacity in non-treated APP/PS1 mice (previously published data)." The reference of the previous data is necessary.

Response: The reference of the mentioned previously published data was included in the manuscript.

(d) The method of Topiramate administration (line 243, Treatment with Topiramate).

Although Topiramate seems to be administered by orally, it should be described. The detail how it was orally administered should also be described.

Response: Topiramate was administered by gavage. This important information was included in the methodology accordingly. Please see line 251.

(e) Animal care. (line 232, 4,1 Animals). 

The admittance of the animal experiments by the institutional animal care committee or the description of the compliance to the general guidelines of the care and use of laboratory animals is necessary.

Response: We thank the reviewer for this observation. The information was updated in the manuscript accordingly. Please see line 245.

(f) Title should include "Alzheimer’s disease".

APP/PS1 mice is generally regarded as Alzheimer model mice. One suggestion of the title is as follows. "Amelioration of behavioral impairments and neuropathology by antiepileptic drug Topiramate in a transgenic Alzheimer’s disease model mice, APP-PS1.

"cerebral amyloidosis" may include vascular amyloidosis.

Response: APP/PS1 is known to be a transgenic mouse model of cerebral amyloidosis. However, as the reviewer’s stated, it is generally regarded as Alzheimer’s disease mouse model. Therefore, we welcome the reviewer’s suggestion and modified the title.

We have addressed all comments of the reviewers and are looking forward to a new consideration of this manuscript.

Yours sincerely,

Dr. Brice Ayissi Owona

Institute of Pathology and Neuropathology

University of Tuebingen

Calwer Str. 3  

72076 Tuebingen, Germany

Reviewer 2 Report

Review of the manuscript “Amelioration of behavioral impairments and   neuropathology by antiepileptic drug Topiramate in a transgenic mice model of cerebral amyloidosis” by Brice Ayissi Owona and coauthors submitted to the “International Journal of Medical Sciences”, MDPI.

Alzheimer’s disease (AD) is a severe neurological disorder, being a main cause of dementia. In 2014, about 5 million Americans suffered from Alzheimer’s disease, and still there is no effective disease- modifying treatment of this disorder. The disease is accompanied by impairments in cognitive ability and social interaction which are caused by neurotoxicity and neuroinflammation. Topiramate is an antiepileptic drug approved by FDA which, according to preliminary data, may ameliorate symptoms of Alzheimer’s disease pathology in combination with levetiracetam. In the current manuscript the authors describe the results of topiramate effect on social behavior and other symptoms of Alzheimer’s disease pathology. This is an important study, the results of which will be interesting for the readership of the journal. 

The following corrections should be made:

Introduction:

Lines 35-36.

“For instance, the anticonvulsant and mood-stabilizing agent Valproic acid (VPA) has been widely used for the treatment of epilepsy”.  After this sentence the authors should add the following sentence and corresponding reference: ”Many other potential antiepileptic agent have been tested acting on various mechanisms of epilepsy. [ref. “New putative epigenetic mechanism of epilepsy”. Frontiers in Neurology, 2017, 8:3. doi: 10.3389/fneur.2017.00003].

Line 43.

“social interaction ability associated cerebral amyloidosis”.  This phrase should be corrected as follows:

“social interaction ability associated with cerebral amyloidosis”. 

Line 51:”…we used topiramate alone in APP/PS1-21 transgenic mouse model in a reduced amount of time and… “should be corrected as follows: ”…we used for a limited time topiramate alone in APP/PS1-21 transgenic mouse model and…”

Results:

Lines 65-70. It is not clear why the authors placed Figure 3 before Figure 2. Should change the order of appearance or numeration according to commonly used standards.

Line 73. “2.2. Effect of Topiramate on treatment on behavioral impairment (nest construction assay)” The sense of this heading is not clear. May be the authors want to say “2.2. Effect of Topiramate treatment on behavioral impairment (nest construction assay)”?

Line 95:”was observed after after 21 days treatment (P<0.05).” Should be corrected as follows: “was observed after 21 days of treatment (P<0.05).”   

Figure 5 legend, line 126. “Representative micro images show the changes in Aβ deposition and microglial activation following …”. It is not clear what the authors mean saying “Representative micro images”. The authors should give an idea how “representative” were selected? Do the authors presented the best out of 3? Out of 10? How these “representative” were different from others?

Discussion:

The authors should speculate how the positive results on behavioral impairments that they found in mice may be expressed in patients with Alzheimer’s disease.

Author Response

Response to reviewers

Manuscript IJMS- 50612

Dear Dr Josephine Xu and Editors,

Thank you very much for consideration of a new version of our manuscript entitled " Amelioration of behavioral impairments and neuropathology by antiepileptic drug Topiramate in a transgenic mouse model of cerebral amyloidosis". We would also like to thank the reviewers for the many helpful comments and suggestions.

In the following, we would like to give a point-by-point response to the comments of the reviewers. In the revised manuscript, we have also marked all the changes.

Reviewer #2:

(1) “For instance, the anticonvulsant and mood-stabilizing agent Valproic acid (VPA) has been widely used for the treatment of epilepsy”.  After this sentence the authors should add the following sentence and corresponding reference:” Many other potential antiepileptic agents have been tested acting on various mechanisms of epilepsy. [ref. “New putative epigenetic mechanism of epilepsy”. Frontiers in Neurology, 2017, 8:3. doi: 10.3389/fneur.2017.00003].

Response: We thank the reviewer for this observation and the mentioned sentence was added.

(2) Line 43.

“social interaction ability associated cerebral amyloidosis”.  This phrase should be corrected as follows:

“social interaction ability associated with cerebral amyloidosis”. 

Response: The correction was made in the text.

(3) Line 51:” we used topiramate alone in APP/PS1-21 transgenic mouse model in a reduced amount of time and… “should be corrected as follows:”…we used for a limited time topiramate alone in APP/PS1-21 transgenic mouse model and…”

Response: The correction was made as requested.

(4) Lines 65-70. It is not clear why the authors placed Figure 3 before Figure 2. Should change the order of appearance or numeration according to commonly used standards.

Response: We thank the reviewer for this observation. The error may have occurred during the editing process. However, the numbering has been exchanged in the manuscript.

(5) Line 73. “2.2. Effect of Topiramate on treatment on behavioral impairment (nest construction assay)” The sense of this heading is not clear. May be the authors want to say “2.2. Effect of Topiramate treatment on behavioral impairment (nest construction assay)”?

Response: The correction was made in the text.

(6) Line 95:” was observed after after 21 days treatment (P<0.05).” Should be corrected as follows: “was observed after 21 days of treatment (P<0.05).”   

Response: The correction was made in the text.

(7) Figure 5 legend, line 126. “Representative micro images show the changes in Aβ deposition and microglial activation following …”. It is not clear what the authors mean saying “Representative micro images”. The authors should give an idea how “representative” were selected? Do the authors presented the best out of 3? Out of 10? How these “representative” were different from others?

Response: We thank the reviewer for this question. The representative images were selected among the images obtained from different areas of the cortex and the hippocampus. Images were taken from all the mice brains, and the best pictures out of 10 images from the same treatment were represented.

(8) Discussion:

The authors should speculate how the positive results on behavioral impairments that they found in mice may be expressed in patients with Alzheimer’s disease.

Response:

We have addressed all comments of the reviewers and are looking forward to a new consideration of this manuscript.

Yours sincerely,

Dr. Brice Ayissi Owona

Institute of Pathology and Neuropathology

University of Tuebingen

72076 Tuebingen, Germany
